# OpenReview forum: "Refusal Direction is Universal Across Safety-Aligned Languages"
_NeurIPS.cc/2025/Conference — NeurIPS 2025 poster_

### Official Review · Reviewer_wcnc · 2025-06-15

**Clarity:** 3
**Significance:** 2
**Originality:** 2
**Rating:** 4
**Confidence:** 5

**Summary:**

The paper investigates whether representations of refusal are language dependent or language agnostic. In order to investigate this question, the authors translate existing datasets of harmful requests into a wide range of languages, resulting in a new multi-lingual dataset called "PolyRefuse". The authors find that refusal directions extracted from any source language can transfer to manipulating refusal behavior in any target language, suggesting that the a model's "refusal direction" is language agnostic. Additionally, they find that refusal directions extracted using different languages converge to the same universal refusal direction.

**Questions:**

- Harmful vs harmless representation analysis
  - Why is the analysis of harmful vs harmless at the same layer and token position as refusal vs non-refusal? Would we expect these concepts to be represented at precisely the same location? Observing a lack in separability at a single activation site does not imply that the model does not cleanly separate these concepts generally.
  - Why use Silhouette Score to measure separability between harmful and harmless requests? My instinct would be to use simple linear probing to measure this. It would also be simple to perform linear probing and sweep over layers and token positions, to address my previous point about locality.
  - Additionally, it is hard to know how to interpret the Silhouette Score. For example, what does a score of 0.4960 mean? Is it good or bad? I find it strange that "de" has a lower Silhouette Score than "yo", despite "de" refusing more harmful requests than "yo" - how do you explain this?
- How does Wildguard work? Why was Wildguard chosen for evaluating compliance vs refusal?
- In figure 4, why display multiple token positions and layers? In my opinion, the point would be much clearer if you only compared the particular activation site (the specific layer and token position from which the refusal direction was extracted), and show that the extracted vectors have very high cosine similarity (although this would then require some baselines to get a sense of what an "unusually large" cosine similarity is in the given dimensionality, but this should be straightforward). My point here is that most of the information displayed in the plot is uninteresting / unimportant (the pair-wise comparisons between vectors at different activation sites) - the important information is concentrated at the comparisons of between vectors from the *same* activation site, and the figure can be greatly simplified to emphasize this.

**Ethical Concerns:**

["NO or VERY MINOR ethics concerns only"]

**Final Justification:**

The core finding that refusal directions are universal across languages is strong. While the geometric analysis has methodological limitations, I think the core result is strong enough to warrant acceptance, although I'm unsure.

**Limitations:**

- I think the most significant limitations are those concerning the separation between harmful and harmless embeddings (see Questions section for details). I think these limitations should be acknowledged, or that there should be additional, more thorough analysis conducted on this important claim.

**Quality:**

2

**Strengths And Weaknesses:**

Strengths:
- The core result of the refusal direction being language agnostic is clear and well-supported by evidence.
- The new dataset seems useful for future research on multi-lingual adversarial robustness.

Weaknesses:
- I think the claims of harmful/harmless separability are weak (see details in Questions section).

---

> ### Author Rebuttal · Authors · 2025-07-31
>
> Dear Reviewer wcnc,
>
> Thank you for your valuable feedback. We appreciate your recognition that our refusal vector being language agnostic is well supported. We want to address your concern on the harmful/harmless separation below:
>
> 1. > Why is the analysis of harmful vs harmless at the same layer and token position as refusal vs non-refusal? Would we expect these concepts to be represented at precisely the same location? Observing a lack in separability at a single activation site does not imply that the model does not cleanly separate these concepts generally.
>
> In Figure 3, we focus on analyzing the model's refusal behavior in different scenarios, rather than probing the harmfulness / harmlessness of the input. Our goal is to understand what happens in the model’s internal representation when it fails to refuse harmful prompts across languages. We focus on the same layer used for extracting refusal directions, as this is where refusal behavior is most directly encoded. We observe that, in such failure cases, the models are usually unable to distinguish harmful inputs from harmless ones at this layer. While harmfulness and refusal are not necessarily identical in how LLMs represent them, our analysis centers on the point where refusal decisions are made. **Since the model needs to first identify harmfulness concept before triggering the refusal behavior, our refusal vector analysis serves as a ‘downstream’ analysis of the ‘upstream’ harmfulness identification event .** We agree that our approach is not a direct analysis but we think this is an interesting perspective to understand the model’s harmfulness concept. In our revised version, we will acknowledge the limitation of indirectness in our discussion to avoid any confusion from the reader.
>
> 2. > Why use Silhouette Score to measure separability between harmful and harmless requests? My instinct would be to use simple linear probing to measure this. It would also be simple to perform linear probing and sweep over layers and token positions, to address my previous point about locality.
>
> We agree that linear probing is a reasonable way to measure separability between harmful and harmless queries. In our study, we chose the Silhouette Score because it is simpler to apply, fully model-agnostic, and directly quantifies the clustering quality of internal representations. It is particularly well-suited for visualizing and comparing separation across languages without introducing additional training components. While linear probing could offer complementary insights, we found the Silhouette Score sufficient to reveal the weaker harmful–harmless separation in non-English settings.
>
> As for the locality concern, **we again want to highlight that our analysis centers around the refusal behaviour and uses refusal behavior differences to infer the model's ability to identify harmfulness.** We appreciate the linear probing approach proposed by the reviewer which we think is very promising for direct harmfulness concept analysis and understanding its relationship with the refusal behavior. As our work focuses on the refusal behavior itself, we want to provide refusal behavior level perspective to understand the harmful/harmless-ness separation, and we leave the investigation of the direct link between refusal and harmfulness to future work.
>
> 3. > Additionally, it is hard to know how to interpret the Silhouette Score. For example, what does a score of 0.4960 mean? Is it good or bad? I find it strange that "de" has a lower Silhouette Score than "yo", despite "de" refusing more harmful requests than "yo" - how do you explain this?
>
> The Silhouette Score assesses clustering quality by combining two factors: (1) cluster compactness—how closely related a point is to others within the same cluster; and (2) between-cluster separation—how well the clusters are distinguished from each other. In our setting, we measure how well the representations of harmful and harmless queries form distinct clusters.
>
> English consistently yields the highest Silhouette Score, as harmful and harmless representations are both tightly clustered and well-separated. In contrast, for other languages, some harmful representations—especially jailbroken ones where the model fails to refuse—deviate from the expected cluster, reducing the score.
>
> For yo (Yoruba), as discussed in Section 4, the model exhibits a lack of refusal capability, and we classify it as safety-misaligned. In Figure 3, the harmful representations in yo are relatively compact, which improves within-cluster compactness. However, they are not well-separated from harmless representations, leading to reduced between-cluster separation. These opposing factors result in a moderate Silhouette Score—occasionally even higher than that of better-aligned languages like de (German).
>
> Ultimately, our aim is to highlight the consistent gap between English and non-English languages, which supports our claim that insufficient harmful–harmless separation contributes to jailbreaks in non-English settings.
>
> We will clarify this point in the revised version of our paper.
>
> 4. > How does Wildguard work? Why was Wildguard chosen for evaluating compliance vs refusal?
>
> The WildGuard [1] is a common and popular LLM classifier that is designed to detect if the model is refusing the question or not. It is more robust than traditional staring matching approaches which relies on only a small set of refusal patterns. It is also considered to be better than LlamaGuard, which is also a LLM classifier, since the WildGuard was specifically optimized to be robust in out of distribution detection scenarios. Therefore, we choose WildGuard as our refusal and compliance detector.
>
> 5. > In figure 4, why display multiple token positions and layers? The important information is concentrated at the comparisons of between vectors from the same activation site, and the figure can be greatly simplified to emphasize this.
>
> We appreciate the reviewer's suggestion to make our figure compact and simple. In Figure 4, we include other position similarity as a comparison to highlight the high similarity at the refusal extraction position. We want to highlight that the high similarity ‘only’ happens at the same token and layer positions, across different languages. We will consider the reviewer’s suggestions to reduce the number of languages and the image size to simplify the figure for better readability.
>
> We thank the reviewer again for the valuable feedback and efforts in reviewing the paper. Please let use know if our explanation address your concerns and we are open to answer any further questions.
>
>
>     [1] Han, Seungju, et al. "WILDGUARD: open one-stop moderation tools for safety risks, jailbreaks, and refusals of LLMs." Proceedings of the 38th International Conference on Neural Information Processing Systems. 2024.

---

> > ### Comment · Reviewer_wcnc · 2025-08-02
> >
> > Thank you for addressing my questions. Based on your responses, I will raise my score. Please include the discussed clarifications in the final manuscript.

---

> ### Author Response · Authors · 2025-08-02
> **Thank you for your review and raising the score**
>
> Dear Reviewer wcnc,
>
> We are very happy to hear that we have addressed your concerns and the score has been raised. We will include the discussed clarification into the final manuscript. Thank you for your careful review which has helped with our paper a lot.

---

### Official Review · Reviewer_rGPS · 2025-06-28

**Clarity:** 3
**Significance:** 3
**Originality:** 3
**Rating:** 4
**Confidence:** 4

**Summary:**

This paper evaluates the cross-lingual transferability of refusal vectors in LLMs. The authors construct a comprehensive evaluation dataset covering 14 high-resource and low-resource languages, with experimental results confirming the significant transferability of refusal vectors across safety-aligned languages.

**Questions:**

- Does the experiment in 5.2 subtract the refusal vectors at all layers?
- The color of Figure 2 can be adjusted, as it currently looks too dark.

**Ethical Concerns:**

["NO or VERY MINOR ethics concerns only"]

**Final Justification:**

Most of my concerns have been addressed and I will keep my borderline accept rating.

**Limitations:**

yes

**Quality:**

3

**Strengths And Weaknesses:**

## Strength

The motivation of this paper is great. While existing LLM safety research mainly focuses on English, this work provides a systematic investigation of refusal vectors in multilingual scenarios, offering valuable insights to the community. The evaluation covers 14 languages, including comparative analyses between high- and low-resource languages. I also enjoy reading the paper, as it is well-structured and easy to follow.

## Weakness

- The current heuristic approach for selecting high-probability refusal tokens (e.g., Table 4) lacks sufficient coverage (e.g., English only includes "I" while there are also many typical refusal phrases like "as" or "sorry"). Therefore, I think the robustness of this method is unknown, and the author should strengthen the discussion in this regard. In my opinion, a simpler method to calculate the "refusal score" may be to use SCAV [1]. It trains a linear classifier, so the classification accuracy on test set of each language can be used to select the most effective refusal vector.

- When constructing the dataset, I really appreciate that the authors evaluated the translation quality. However, since there are cultural differences in language, translation may amplify certain biases, so it is necessary to manually evaluate the quality of the dataset.

- In section 5.2, the author concluded that the refusal vectors obtained from English can be effectively transferred to other languages. I want to see what happens if the refusal vectors are used for defense. For example, when applying English refusal vectors to  low-resource languages, will it make the model safer?

- Does the found phenomenon also exist in the reasoning model (like Qwen3 or DeepSeek-R1)/ MOE architecture model?

- Related work should be supplemented with more papers explaining LLM security mechanisms, such as [1][2].

## Reference
[1] Xu, Zhihao, et al. "Uncovering safety risks of large language models through concept activation vector." Advances in Neural Information Processing Systems 37 (2024): 116743-116782.

[2] Li, Tianlong, et al. "Revisiting Jailbreaking for Large Language Models: A Representation Engineering Perspective." Proceedings of the 31st International Conference on Computational Linguistics. 2025.

---

> ### Author Rebuttal · Authors · 2025-07-31
>
> Dear Reviewer rGPS,
>
> Thanks for your valuable feedback and suggestions. We address your concern below:
>
> 1. > The current approach for selecting refusal tokens lacks sufficient coverage. The author should strengthen the discussion in this regard.
>
> We understand reviewers’ concern that the tokens we consider may not cover all the usual refusal tokens. Tokens like ‘As’ or ‘Sorry’ are not included in all the models. However, we ensure the coverage by first calculating the frequency of this token appears as the first token when refusing the query and ‘‘as’ and ‘sorry’ do not appear as a common first refusal token. We see that the common refusal tokens are highly model dependent. We show top 4 tokens statistics below for the tokens for Llama3.1-8B answering 572 unsafe queries:
>
> |  | 'l' | 'Here' | 'If' | '**' |
> | :--- | :--- | :--- | :--- | :--- |
> | Count | 555 | 3 | 2 | 2 |
>
> We agree that such heuristic approach may not be ideal, but we are impressed by its effectiveness and efficiency. We are grateful for the great suggestion on using more accurate selection method like SCAV, and we think such method could be used for more complex refusal mechanism feature extraction in reasoning model which we believe is an important future direction.
>
> 2. > When constructing the dataset, I really appreciate that the authors evaluated the translation quality. However, since there are cultural differences in language, translation may amplify certain biases, so it is necessary to manually evaluate the quality of the dataset.
>
> Thank you for the comment. We acknowledge the concern regarding cultural differences and potential bias amplification through translation. In our case, the harmful vs. harmless distinction (e.g., “how to make a bomb”) remains largely consistent across languages, which is why we did not conduct a full manual inspection. That said, we included several representative examples in the appendix, and they show that cultural differences did not result in flips or changes in the harmfulness of the questions/answers after translation. Furthermore, the harmfulness features in the dataset are culture-agnostic (eg. bio-weapon, cyber attack, spreading misinformation) based on our initial manual inspect, which addresses the concern of cultural bias.
>
> 3. > I want to see what happens if the refusal vectors are used for defense. For example, when applying English refusal vectors to low-resource languages, will it make the model safer?
>
> Yes, applying the refusal vector to low-resource languages will increase the refusal rate. Below is the refusal rate of Llama3.1-8B-Inst in different languages when we do the addition operation. We see that the refusal rate increases across all languages. The gain is especially obvious when applying in low resource languages. However, such addition will also increase the refusal rate when answering neutral queries – an observation shown in [1]. Therefore, the current adding refusal vector is currently not an optimal solution for defending cross-lingual jailbreak, which we believe is an important future direction.
>
> |  | En | Ar | It | De | Es | Fr | Th | Ja | Yo |
> | :--- | :--- | :--- | :--- | :--- | :--- | :--- | :--- | :--- | :--- |
> | Before addition | 0.98 | 0.94 | 0.98 | 0.97 | 0.98 | 0.98 |  0.90 | 0.75 | 0.17 |
> |After addition | 1.0 | 1.0 | 1.0 | 1.0 | 1.0 | 1.0  | 1.0 | 0.93 | 0.79 |
>
> 4. > Does the found phenomenon also exist in the reasoning model (like Qwen3 or DeepSeek-R1)/ MOE architecture model?
>
> Currently the refusal direction identification line of work focuses on non-reasoning dense models. Recent study has shown that the test-time scaling can improve the robustness of the safety alignment, which indicates that the reasoning model may have a more complex mechanism in terms of refusing to harmful queries [2][3], which may increase the difficulty of the refusal activation vector extraction. We cautiously position broader generalization to reasoning/MoE models as a priority for follow-up research.
>
> 5. >Related work should be supplemented with more papers explaining LLM security mechanisms, such as [1][2].
>
> Thank you very much for providing the papers which helps our paper a lot for clarity. We will reference the papers in the revised version.
>
>     [1] Arditi, Andy, et al. "Refusal in Language Models Is Mediated by a Single Direction." The Thirty-eighth Annual Conference on Neural Information Processing Systems.
>     [2] Guan, Melody Y., et al. "Deliberative alignment: Reasoning enables safer language models." arXiv preprint arXiv:2412.16339 (2024).
>     [3] Kim, Taeyoun, et al. "Reasoning as an Adaptive Defense for Safety." arXiv preprint arXiv:2507.00971 (2025).

---

> > ### Comment · Reviewer_rGPS · 2025-08-02
> >
> > Thanks to the authors for their responses. I will keep my score.

---

> ### Author Response · Authors · 2025-08-02
> **Thank you for your review**
>
> Dear Reviewer rGPS,
>
> Thanks for keeping a positive score for our paper. We are grateful for your effort in giving feedback and thoughtful suggestions from which our paper benefits a lot.

---

### Official Review · Reviewer_h65C · 2025-07-02

**Clarity:** 3
**Significance:** 2
**Originality:** 3
**Rating:** 4
**Confidence:** 4

**Summary:**

This paper claims that the "refusal direction", i.e., the vector obtained as a function of the differences between the activation values of an LLM when refusing harmful prompts vs responding to harmless prompts, works 'universally' across languages. Prior work has shown the impact of this refusal direction vector in controlling how often the model refuses harmful queries for English. This work extends this analysis to multiple languages - (1) the authors show that the refusal direction obtained from english can be used to get the model to refuse less when prompted with harmful prompts in other languages, and (2) the refusal direction obtained from different languages can also be used to get the model to refuse less in other languages. The authors also carry out a geometric analysis of activation vectors and refusal directions under different scenarios to support their claims.

**Questions:**

L237: "at the refusal extraction layer" -- what does this mean?

**Ethical Concerns:**

["NO or VERY MINOR ethics concerns only"]

**Final Justification:**

The main weakness I pointed out was a significant lack of control experiments in the paper to support the main claims. The authors have carried out preliminary control experiments and I find the results encouraging, so I am changing my score to acceptance.

**Limitations:**

The authors discuss some limitations of their work that are not immediately addressed but could be looked at in future research.

**Quality:**

3

**Strengths And Weaknesses:**

Strengths:

- The motivation for this paper is strong: for any alignment phenomenon demonstrated in English, it's great to get insights on how it holds up for other languages. In general it is known to some extent that multilinguality can be exploited to jailbreak LLMs and so such studies that can (potentially) guide solutions to solve such problems are interesting for the community.
- The Polyrefuse dataset, which includes 14 different languages, can be a useful artifact for future research.

Weaknesses:

- My main issue is that I am not convinced that the claim of universality is adequately supported. This stems from lack of experiments in two parts:

(1) There is no control: For the experiments in sections 5.2-5.3, there should be results reported with some control vectors/directions. I would have expected to see the effect of using random vectors instead of the refusal direction calculated by the method. I would also have expected another control for the data - if you used some other non-harmful prompts "neutral" dataset (say open-instruct) instead of the harmful dataset (i.e., you wouldn't be calculating refusal safety direction anymore), what happens? ---- Basically my concern is that when you are 'ablating' these refusal vectors from the LLM and showing decrease in refusal, how do we know if it is because of these refusal vectors and not because you are in some sense damaging the LLM?

(2) No experiments showing increase of refusal: The paper does not carry out experiments to show that when the refusal direction is added (eq. 5), it increases refusal rate. For a strong claim like universality, in my opinion, the authors need to show this in both settings of sections 5.2-5.3.

- Minor point: I am also not very convinced by the geometric analysis. For instance in Figure 3, I would think that the refusal direction for non-English languages would be from the centroid of "harmless_<lang>" to centroid of "harmful_refuse_<lang>". But the dotted arrows simply indicate the parallel line to the english counterpart which can be misleading.

---

> ### Author Rebuttal · Authors · 2025-07-31
>
> Dear Reviewer h65c,
>
> Thank you very much for your thoughtful review and valuable feedback. We address your concern below:
>
> 1. > Basically my concern is that when you are 'ablating' these refusal vectors from the LLM and showing decrease in refusal, how do we know if it is because of these refusal vectors and not because you are in some sense damaging the LLM?
>
> We understand your concern that the decrease in refusal may be due to the vector ablation damaging the model. However, the answer to this concern is: No, the vector ablation is not damaging the model.
> **Also, this is the core design of the refusal vector extraction method in the original paper: the ablation operation should only control the refusal, not damaging any other aspects of the model.** This is done by strictly filtering our vectors that change the KL on the first token probability over 0.1. The original paper did extensive experiments on demonstrating the surgical effect of the vector, **including comparing it to different vectors, and doing the addition operation.** We didn’t show the same experiments in our paper as we thought it was shown fairly well in the original and other follow-up papers [1][2][3]. However, we fully understand your concern and would like to add such experiments to re-confirm that the vector ablation is not damaging the model, and any other vectors will do.
>
> **Control1: What happens if we ablate a different vector?**
>
> Here we not only show the model’s refusal on harmful query data, but also general datasets that evaluate the LLM capability, such as MMLU, ARC-R, and PPL on Wikitext. We compare the refusal vector we extracted to two random vectors which are chosen by randomly selecting a layer and token position from the candidate diff-in-means vectors.
>
> | | Refusal Rate | MMLU | PPL | ARC-C |
> | --- | :--- | :--- | :--- | :--- |
> | Before ablation | 0.99 | 68.5 | 8.65 | 52.4 |
> | Ablate Refusal Vector | 0.02 | 68.0 | 8.71 | 52.5 |
> | Ablate Random vector 1, layer 12, token position 3 | 1.0 | 65.8 | 9.17 | 49.6 |
> | Ablate Random vector 2, layer 3, token position 3 | 0.55 | 66.0 | 9.21 | 32.8 |
>
> We see that, ablating the refusal vector can successfully remove the refusal behavior on the harmful dataset, while keeping the general capabilities. When we removing a random diff-in-means vector, we see that the model is damaged in terms of general capability, and it’s refusal score is not completely removed. We can see that the random vector 2 is indeed **damaging** the model and resulting a refusal rate decrease. However, our true refusal vector is only ablating the refusal behavior not others. We also recommend reviewer to check the section 4.3 in the original refusal vector extraction paper[1], for more details and discussion on the surgical effect of the vector ablation. **We want to highlight that such refusal extraction method is a well established work which was shown to be very effective in surgical manipulation of the model.**
> For qualitative result, we refer the reviewer to the Appendix Section B in our submission where we show that the model’s generation quality is not impacted across the languages after ablation.
>
> **Control2: Increase of Refusal**
>
> Increasing the refusal by applying addition operation has been shown to be effective in prior works [1][2][3]. In the original refusal vector extraction paper, the author successfully increased the refusal rate of the models without damaging the general capabilities.
> Given the strong transferability of the ablation operation, we expect the same trend across different languages. To further validate this, we test the addition operation across languages on Llama3.1-8B-Instruct and we see a consistent refusal increase. Thanks again for the valuable suggestions, and we plan to include such addition operations in our revised version for addressing any potential concerns.
>
> |  | En | Ar | It | De | Es | Fr  | Th | Ja | Yo |
> | :--- | :--- | :--- | :--- | :---  | :--- | :--- | :--- | :--- | :--- |
> | Before addition | 0.98 | 0.94 | 0.98 | 0.97 | 0.98 | 0.98 | 0.90 | 0.75 | 0.17 |
> | After addition | 1.0 | 1.0 | 1.0 | 1.0 | 1.0 | 1.0 | 1.0 | 0.93 | 0.79 |
>
> 2. > Geometric analysis: refusal direction for non-english languages would be from  the harmless to harmful centroid.  But the dotted arrows simply indicated the parallel line to the English counterpart
>
> Thank you for the comment. These lines are intended to emphasize that the refusal directions across languages are approximately parallel, a key observation we make in our analysis. While they may not be perfectly parallel in all cases, they serve to illustrate the general trend. This approximate parallelism is also supported by the quantitative results in Figure 4, where refusal directions across languages exhibit consistently high cosine similarity. We will clarify in the revised version that these vectors are illustrative and not strictly drawn from centroids, to avoid potential misunderstanding.
>
> 3. > L237: "at the refusal extraction layer" -- what does this mean?
>
> The refusal extraction layer means the layer at which we extract the refusal vector.
>
> We thank again for the valuable feedback from the reviewer and would like to answer any questions if concern remains. We appreciate your dedicated efforts on reviewing the paper which helps improve the quality of the paper a lot. We are grateful if the reviewer finds our explanation satisfactory and consider re-evaluating the paper.
>
>     [1] Arditi, Andy, et al. "Refusal in Language Models Is Mediated by a Single Direction." The Thirty-eighth Annual Conference on Neural Information Processing Systems.
>     [2] Wang, Xinpeng, et al. "Surgical, Cheap, and Flexible: Mitigating False Refusal in Language Models via Single Vector Ablation." The Thirteenth International Conference on Learning Representations.
>     [3] Wollschläger, Tom, et al. "The Geometry of Refusal in Large Language Models: Concept Cones and Representational Independence." Forty-second International Conference on Machine Learning

---

> > ### Comment · Reviewer_h65C · 2025-08-05
> > **Response**
> >
> > Thanks for your response and additional experiments. I understand that the effectiveness of the surgical manipulation of the method has been shown in prior works, but unless I am mistaken, the setting and data of application in your paper is different, which warrants carrying out extensive control experiments to support your central claims of universality of refusal direction.
> > I am encouraged by your results for both the controls, and hence am increasing my score to recommend acceptance. I strongly recommend the authors to carry out these control experiments a bit more rigorously (e.g., average across multiple random vectors from different layers etc.) and discuss them at length in the final version of the paper.

---

> ### Author Response · Authors · 2025-08-05
>
> Dear Reviewer h65C,
>
> Thank you for your suggestions and raising your score to recommend acceptance. Given the additional page, we are happy to include the control experiment results in the main paper. We do have the ablation results on fully sweeping across the layers and the token positions, which was done similarly as in the prior works and we will also include this detailed result in the appendix. We agree that such experiments complement our current results and can help address potential concerns from the readers. Thank you for helping review the paper!

---

### Official Review · Reviewer_iBP3 · 2025-07-05

**Clarity:** 3
**Significance:** 3
**Originality:** 3
**Rating:** 4
**Confidence:** 4

**Summary:**

This work shows that refusal direction is universal across several languages. Refusal directions extracted from English generalize to other languages, and refusal direction from any safety-aligned language applies to others as well. The authors further highlight that despite the universality of refusal direction, some languages are more vulnerable to jailbreak attacks as the harmful and harmless representations are less separable in these languages.

**Questions:**

Please refer to the weaknesses above.

**Ethical Concerns:**

["NO or VERY MINOR ethics concerns only"]

**Final Justification:**

The rebuttal addresses several concerns raised. I am inclined towards acceptance and maintain my rating of "Borderline Accept". The paper would benefit from more in-depth study on why the generalization of refusal direction does not hold for a few languages, and some hints on how this can be improved.

**Limitations:**

Yes

**Quality:**

3

**Strengths And Weaknesses:**

**Strengths**
- The two observations highlighted in the paper are insightful and useful for understanding safety concerns in low resourced languages
- The observations in this work could motivate methods to defend against cross-lingual jailbreaks as well
- Contribution of a dataset PolyRefuse with harmful prompts across 14 languages
- Thorough experimentation with intuitive visualizations

**Weaknesses/ Areas for further improvement**:
 - Understanding scaling laws with respect to this universality would be very useful as this trend may change across model sizes - how does this behaviour change for smaller / larger models
 - It is not clear what types of prompts this observation generalizes to - is it natural unsafe prompts or adversarial jailbreaks - an analysis on transferability for different types of unsafe/ jailbreak prompts would be insightful
 - The datasets have been generated by translating from English. It would be useful to also check if the transferability holds on completely independent datasets (maybe this is already achieved as the authors maintain a test split)
 - The reason why it is hard to separate harmful and harmless prompts in some languages is not clear - is it because the model is unable to understand the words, or is there a lack of generalization of safety concepts - more experiments / analysis on this would be helpful. For example, if the same target model is asked to translate the two prompts back to English, are the translated prompts correct and well distinguishable?

---

> ### Author Rebuttal · Authors · 2025-07-31
>
> Dear Reviewer iBP3,
>
> Thank you for your valuable feedback. We appreciate that you find our work insightful and useful for understanding the safety concern in low resource languages, and our work can motivate methods for defend cross-lingual jailbreaks. We want to address you concern below.
>
> 1. > Understanding the scaling laws with respect to universality would be useful.
>
>
> We appreciate the reviewer’s suggestion to explore scaling trends in refusal vector universality. In our experiment in Figure 1, we include models ranging from 2B to 70B parameters across multiple families. We consistently observe the transferability of the refusal vector across all model sizes. Due to computational constraints, we are currently unable to extend our experiments to larger models or increase the granularity of scaling analysis. However, we believe our current results already provide strong evidence that the observed universality is consistent across model scales.
>
>  2. > It is not clear to what type of prompts the observation generalizes to - is it natural unsafe prompts or adversarial jailbreaks - an analysis on transferability for different types of unsafe/ jailbreak prompts would be insightful.
>
> In our study, we focus on natural unsafe prompts because this is where the refusal mechanism in language models is designed for. Currently, the refusal mechanism in LLM is only designed to defend against natural unsafe prompts, and we are interested in the transferability of such behavior across languages. In the adversarial setting, with the advantage of using optimization techniques or access to model weights the jailbreak prompt can always bypass the refusal mechanism, even if we can transfer the refusal in the first place. We acknowledge the complexity of the adversarial setting due to its dynamic nature, and we believe this is an important future direction which requires careful investigation.
>
> As within the natural unsafe prompt, we do include different prompt styles by using different data sources in training and the test set. We use data from Advbench, MaliciousInstruct and TDC2023, which include question and instruction prompt styles. For generalizing to a different dataset, please see response below.
>
> 3. > The datasets have been generated by translating from English. It would be useful to also check if the transferability holds on completely independent datasets.
>
> We appreciate the reviewer's concern on the transferability to independent data. First, we want to highlight that the generalization of the refusal vector to unseen and independent data was shown to be very effective in previous works [1][3]. For example, in [1] the author tested the model refusal rate on a different dataset JailbreakBench [2] which is different from the data used for vector extraction. To further validate this, we test on a multilingual version of the JailbreakBench, which was not used at all in our method. We show the refusal rate of Gemma2-9B-It before and after the ablation below:
> |                      | En   | De   | Es   | Fr   | It   | Nl   |
> | :------------------- | :--- | :--- | :--- | :--- | :--- | :--- |
> | Before Ablation | 0.99 | 0.92 | 0.95 | 0.97 | 0.99 | 0.97 |
> | After Ablation  | 0.12 | 0.18 | 0.12 | 0.08 | 0.11 | 0.11 |
>
> Such generalization in the cross-lingual setting is within our expectation, given the previous observation in the English setting and our strong transferability result. We appreciate the reviewer’s suggestions on the data independence check.
>
> 4. > The reason why it is hard to separate harmful and harmless prompts in some languages is not clear - is it because the model is unable to understand the words, or is there a lack of generalization of safety concepts - more experiments / analysis on this would be helpful. For example, if the same target model is asked to translate the two prompts back to English, are the translated prompts correct and well distinguishable?
>
> Thank you for your great suggestion and we agree this is an important question to ask. In our observation, the reason is mixed but mostly due to the poor generalization of safety concept. For most languages such as Korean, we see the model has no problem understanding the harmful prompt and actually give harmful responses to fulfill the queries.  However, for very low-resource languages like Yoruba, the model’s response quality is poor, giving no useful information to the harmful queries. However, this is the only language we see where the model has problem giving useful responses. An this is the only language which is not safety aligned across languages. We also refer the reviewer to the model response examples in the Appendix section B, where we show that the model has a good understanding of the prompt and gives harmful responses across the languages.
>
> We thanks again for the valuable feedback from the reviewer and we are happy to answer any addition questions for further concerns.
>
>     [1] Arditi, Andy, et al. "Refusal in Language Models Is Mediated by a Single Direction." The Thirty-eighth Annual Conference on Neural Information Processing Systems.
>     [2] Chao, Patrick, et al. "Jailbreakbench: An open robustness benchmark for jailbreaking large language models." Advances in Neural Information Processing Systems 37 (2024): 55005-55029.
>     [3] Wang, Xinpeng, et al. "Surgical, Cheap, and Flexible: Mitigating False Refusal in Language Models via Single Vector Ablation." The Thirteenth International Conference on Learning Representations.

---

> > ### Comment · Reviewer_iBP3 · 2025-08-04
> >
> > I thank the authors for their rebuttal. I would like to keep the same score as before. The paper would benefit from more in-depth study on why this generalization does not hold for a few languages, and some hints on how this can be improved.

---

> > > ### Author Response · Authors · 2025-08-05
> > >
> > > Dear Reviewer iBP3,
> > >
> > > Thank you for your suggestion on discussing more on the generalization of the finding. Based on our experiments across 14 languages, the only language where we don't see the bi-directional transferability of the refusal direction is Yoruba, which is also the only one language that is not safety-aligned. However, we can indeed increase the refusal in Yoruba by applying the refusal direction extracted from English, as we show in the response to Reviewer h65c and rGPS. This unidirectional transferability makes sense since we can't extract a meaningful refusal vector in a language if that language doesn't encode such refusal behavior.  We are happy to give more detailed discussion and insights in the final version of the paper. Thank you for your advice.

---

### Note · Authors · 2025-08-12

Dear Reviewers and AC,

Thank you for taking your time to review our paper. We are grateful for the overall positive rating based on the reply (Reviewer iBP3 & rGPS kept their positive score of 4, and Reviewer h65C & wcnc raised their scores to recommend acceptance.)

We are happy to have addressed all the points raised by the reviewers including:
_"the control study on the surgical effect of the vector ablation"_, _"why we use refusal vector as a lens to analyse harmful/harmless-ness separation"_, _"effectiveness of increasing refusal in low resource languages by adding refusal vector"_, and _"the discussion on why generalisation is poor on languages that are not safety-aligned"_. We appreciate the valuable feedback, and we will include these points in the final version of the paper.

We also realised that we have missed two minor questions from Reviewer rGPS, which we reply to below:
- Does the experiment in 5.2 subtract the refusal vectors at all layers?
	- We extract a single refusal vector and subtract it across all layers, as we described in 3.2.  We refer the reviewer to [1] for the intuition behind it.
- The color of Figure 2 can be adjusted, as it currently looks too dark.
	- Thank you for your feedback and we will adjust it to a lighter color in the final version.


Thanks again for your review!

Authors
```
[1] Arditi, Andy, et al. "Refusal in Language Models Is Mediated by a Single Direction." The Thirty-eighth Annual Conference on Neural Information Processing Systems.``
```

---

### Decision · Program_Chairs · 2025-09-17

**Decision:**

Accept (poster)

**Comment:**

The paper studies the cross-lingual transferability of "refusal directions" in large language models, showing that these safety mechanisms are largely universal across languages.

Strengths: All reviewers appreciated the strong motivation for extending LLM safety research beyond English to multilingual settings. The PolyRefuse dataset covering 14 languages was consistently praised as a valuable contribution. Reviewers found the core empirical finding compelling: refusal directions extracted from any language can transfer to manipulate safety behavior in other languages, suggesting language-agnostic safety representations.

Limitations: Multiple reviewers initially questioned whether the "universality" claim was adequately supported, specifically noting the lack of control experiments (e.g., using random vectors instead of refusal directions). Reviewers also raised concerns about the heuristic method for selecting refusal tokens and potential limitations from using translated rather than culturally native harmful prompts.

During the rebuttal, the reviewers' concerns were largely addressed. The authors conducted additional control experiments that strengthened the evidence for their claims. While some methodological limitations remain (particularly around the geometric analysis and token selection coverage), reviewers converged on acceptance, viewing the core empirical contribution as sufficiently robust and valuable for the community.